# Towards Feasible Higher-Dimensional Potential Heuristics

**Primary Keywords:** *None*

## Abstract

Potential heuristics assign numerical values (potentials) to state features, where each feature is a conjunction of facts. It was previously shown that the informativeness of potential heuristics can be significantly improved by considering complex features, but computing potentials over all pairs of facts is already too costly in practice. In this paper, we investigate whether using just a few high-dimensional features instead of all conjunctions up to a dimension $n$ can result in improved heuristics while keeping the computational cost at bay. We focus on (a) establishing a framework for studying this kind of potential heuristics, and (b) whether it is reasonable to expect improvement with just a few conjunctions. For (a), we propose two compilations that encode each conjunction explicitly as a new fact so that we can compute potentials over conjunctions in the original task as one-dimensional potentials in the compilation. Regarding (b), we provide evidence that informativeness of potential heuristics can be significantly increased with a small set of conjunctions, and these improvements have positive impact on the number of solved tasks.

## 1 Introduction

In classical optimal planning, potential heuristics (Pommerening et al. 2015; Pommerening, Helmert, and Bonet 2017) are a family of admissible (and consistent) heuristics computed as sums of potentials (numerical values) over state features that are sets of facts (conjunctions). Potentials ensuring admissibility and consistency can be found by solving a certain linear program with any optimization criteria that can be designed to emphasize different properties of the resulting heuristics (Seipp, Pommerening, and Helmert 2015; Fišer, Horčík, and Komenda 2020). It was shown that increasing the size of conjunctions can eventually lead to optimal heuristics (Pommerening, Helmert, and Bonet 2017). Computation of potential heuristics over single facts and pairs of facts is polynomial. However, computing the heuristics over all pairs of facts is already too computationally costly in practice, and for conjunctions of size 3 or more, it becomes coNP-hard to decide heuristic consistency.

Here, we focus on the question whether computing potential heuristics over just a few conjunctions can lead to a significantly more informed heuristic that can be successfully used in practice. We leverage prior work of Steinmetz and Hoffmann (2018) showing that potential heuristics over conjunctions can be computed via the so-called $\Pi^\mathcal{C}$ compilation (Haslum 2012) where conjunctions $\mathcal{C}$ are explicitly represented as facts. Potential heuristics over conjunctions $\mathcal{C}$ can be computed as single-fact potential heuristics in $\Pi^\mathcal{C}$, but the price we pay is the worst-case exponential blow-up in $|\mathcal{C}|$ of the $\Pi^\mathcal{C}$ encoding. Since $\Pi^\mathcal{C}$ does not preserve state space of the original task perfectly and thus does not preserve all (potential) heuristics, we introduce a new compilation $\Pi^\mathcal{C}_{\text{exact}}$ that remedies this pitfall. This allows us to plug-in a set of conjunctions $\mathcal{C}$ as input, and obtain a potential heuristic over $\mathcal{C}$ as output, while exchanging computational complexity for blow-up in the task encoding. Moreover, we use prior work on mutual exclusion state invariants to mitigate the blow-up (Keyder, Hoffmann, and Haslum 2014; Fišer and Komenda 2018; Fišer, Horčík, and Komenda 2020).

To test whether it is possible to increase informativeness of potential heuristics with only few conjunctions, we use a simple greedy uninformed algorithm to obtain improving conjunctions. We show in our experiments that it is, indeed, often the case that a small number of conjunctions leads to better heuristic estimates. We test this approach with explicit-state search, but also with symbolic search where potential heuristics can be used via so-called operator-potential heuristics (Fišer, Torralba, and Hoffmann 2022a,b). We show that even in this simple setting, we are able to increase the number of solved tasks in some domains. We leave the question how to intelligently find the improving conjunctions to future work.

## 2 Background

We consider the finite domain representation (FDR) of planning tasks (Bäckström and Nebel 1995). An **FDR planning task** is a tuple $\Pi = \langle \mathcal{V}, \mathcal{O}, I, G \rangle$. $\mathcal{V}$ is a finite set of **variables**, each $v \in \mathcal{V}$ has a finite **domain** $\text{dom}(v)$. A **fact** $\langle v, x \rangle$ is a pair of a variable $v \in \mathcal{V}$ and one of its values $x \in \text{dom}(V)$. The set of all facts is denoted by $\mathcal{F} = \{\langle v, x \rangle \mid v \in \mathcal{V}, x \in \text{dom}(V)\}$, the set of facts of variable $v$ is denoted by $\mathcal{F}_v = \{\langle v, x \rangle \mid x \in \text{dom}(v)\}$, and similarly for sets of facts $V$: $\mathcal{F}_V = \bigcup_{v \in V} \mathcal{F}_v$. Given $p \subseteq \mathcal{F}$, $\mathcal{V}(p)$ denotes all variables appearing in $p$, i.e., $\mathcal{V}(p) = \{v \mid \langle v, x \rangle \in p\}$, and we use a shorthand $\mathcal{F}_p = \mathcal{F}_{\mathcal{V}(p)}$.

A **partial state** $p \subseteq \mathcal{F}$ is a set of facts s.t. there is at most one fact of each variable, i.e., $|p \cap \mathcal{F}_v| \leq 1$ for every $v \in \mathcal{V}$. $p[v]$ denotes the value assigned to $v \in \mathcal{V}(p)$ in the partial state $p$. A partial state $s$ is called **state** if $|s| = |\mathcal{V}|$. $I$ is an

**initial state**. $G$ is a partial state called **goal**, and a state $s$ is a **goal state** if $G \subseteq s$. A partial state $p \subseteq \mathcal{F}$ is also called a **conjunction**, and we say that a conjunction $c$ is true in the state $s$ if $c \subseteq s$ and we say it is false otherwise. Let $p, t$ be partial states. We say that $t$ **extends** $p$ if $p \subseteq t$.

$\mathcal{O}$ is a finite set of **operators**, $o \in \mathcal{O}$ is defined by its partial states precondition $\mathrm{pre}(o)$ and effect $\mathrm{eff}(o)$, and a cost $\mathrm{cost}(o) \in \mathbb{R}_0^+$. We assume $\mathrm{pre}(o) \cap \mathrm{eff}(o) = \emptyset$. $o \in \mathcal{O}$ is **applicable** in a state $s$ if $\mathrm{pre}(o) \subseteq s$. The **resulting state** of applying an applicable operator $o$ in a state $s$ is another state $o[\![s]\!] = (s \setminus \mathcal{F}_{\mathrm{eff}(o)}) \cup \mathrm{eff}(o)$, i.e., applying $o$ on $s$ changes the values of variables according to the effect $\mathrm{eff}(o)$ and keeps the variables not mentioned in $\mathrm{eff}(o)$ unaffected. We also use $\mathrm{app}(o) = (\mathrm{pre}(o) \setminus \mathcal{F}_{\mathrm{eff}(o)}) \cup \mathrm{eff}(o)$ to denote the partial state resulting from $o$ applied to its own precondition.

A sequence of operators $\pi = \langle o_1, \ldots, o_n \rangle$ is applicable in a state $s_0$ if there are states $s_1, \ldots, s_n$ such that $o_i$ is applicable in $s_{i-1}$ and $s_i = o_i[\![s_{i-1}]\!]$ for $i \in \{1, \ldots, n\}$. The resulting state is $\pi[\![s_0]\!] = s_n$ and $\mathrm{cost}(\pi) = \sum_{i=1}^n \mathrm{cost}(o_i)$ denotes the cost of $\pi$. A sequence of operators $\pi$ is called an $s$-**plan** iff $\pi$ is applicable in a state $s$ and $\pi[\![s]\!]$ is a goal state, $I$-plan is called simply **plan**. An $s$-plan $\pi$ is called **optimal** if its cost is minimal among all $s$-plans. A sequence of operators is called **path** if it is applicable in the initial state.

A state $s$ is **reachable** if there exists an operator sequence $\pi$ applicable in $I$ such that $\pi[\![I]\!] = s$. $\mathcal{R}(\Pi)$ denotes the set of all reachable states in $\Pi$. An operator $o$ is **reachable** if it is applicable in some reachable state. A state $s$ is a **dead-end** if $G \not\subseteq s$ and there is no $s$-plan.

A **heuristic** $h : \mathcal{R}(\Pi) \mapsto \mathbb{R} \cup \{\infty\}$ estimates the cost of optimal $s$-plans. The **optimal heuristic** $h^\star(s)$ maps each reachable state $s$ to the cost of the optimal $s$-plan or to $\infty$ if $s$ is a dead-end. A heuristic $h$ is called (a) **admissible** iff $h(s) \leq h^\star(s)$ for every reachable state $s \in \mathcal{R}(\Pi)$; (b) **goal-aware** iff $h(s) \leq 0$ for every reachable goal state $s$; and (c) **consistent** iff $h(s) \leq h(o[\![s]\!]) + \mathrm{cost}(o)$ for all reachable states $s \in \mathcal{R}(\Pi)$ and operators $o \in \mathcal{O}$ applicable in $s$.

Note that we define heuristics over the reachable states (instead of all states) because we intend to use heuristics in a (forward) heuristic search and because we use state invariants describing the reachable state space for improving the heuristic values (h-values). Also note that we allow negative heuristic values as is usual in literature on potential heuristics (the standard interpretation is that during the search, negative heuristic values are interpreted as zero). It is well-known that goal-aware and consistent heuristics are also admissible.

## 3 Mutexes and Disambiguation

It was previously shown that utilizing mutual exclusion invariants (mutexes) significantly improves informativeness of potential heuristics (Fišer, Horčík, and Komenda 2020). A mutex is a set of facts that is not part of any reachable state, i.e., $M \subseteq \mathcal{F}$ is a **mutex** if $M \not\subseteq s$ for every $s \in \mathcal{R}(\Pi)$.

The most obvious mutex in an FDR task is a set of facts containing two facts of the same variable, but more mutexes can be inferred by the $h^m$ heuristic (Bonet and Geffner 2001; Alcázar and Torralba 2015) or by inference of so called fam-

groups on lifted (Helmert 2009; Fišer 2020, 2023) or ground (Fišer and Komenda 2018) level. Clearly, every superset of a mutex is also a mutex. For notational convenience, we use the notion of a mutex-set.

**Definition 1.** A set of sets of facts $\mathcal{M} \subseteq 2^{\mathcal{F}}$ is called a **mutex-set** if (a) every $M \in \mathcal{M}$ is a mutex, and (b) for every $M \in \mathcal{M}$ and every $f \in \mathcal{F}$ it holds that $M \cup \{f\} \in \mathcal{M}$, and (c) for every variable $v \in \mathcal{V}$ and every pair of facts $f, f' \in \mathcal{F}_v$, $f \neq f'$, it holds that $\{f, f'\} \in \mathcal{M}$.

In other words, a mutex-set is an upper set of a set of mutexes (a,b) and it always contains all mutexes that can be inferred directly from task's variables (c). This allows us to write $p \in \mathcal{M}$ if we want to express that the set of facts $p$ is not a partial state (i.e., it contains two facts of the same variable), or that all states extending $p$ are not reachable. Note also that $p \notin \mathcal{M}$ implies $p$ is a partial state.

Mutexes can also be used for inference of disambiguations (Fišer, Horčík, and Komenda 2020). A disambiguation of a variable $v$ for a partial state $p$ is a set of facts $X \subseteq \mathcal{F}_v$ from the same variable $v$ such that every reachable state extending $p$ contains one fact from $X$. In other words, disambiguation of $v$ for $p$ allows us to filter out facts of the variable $v$ that cannot be part of any reachable state extending $p$.

**Definition 2.** Let $v \in \mathcal{V}$ denote a variable, and let $p$ denote a partial state. $X \subseteq \mathcal{F}_v$ is called a **disambiguation of $v$ for** $p$ if for every $s \in \mathcal{R}(\Pi)$ s.t. $p \subseteq s$ it holds that $X \cap s \neq \emptyset$.

Disambiguations can also be used for finding unreachable operators and determining unsolvability of tasks. If, for some operator $o \in \mathcal{O}$, a disambiguation of some $v \in \mathcal{V}$ for $\mathrm{pre}(o)$ is empty, then $o$ is unreachable; and if a disambiguation of some $v \in \mathcal{V}$ for $G$ is empty, the task is unsolvable. So, from now on, we will consider only tasks for which we have non-empty disambiguations of the goal and operators' preconditions. We use the following **disambiguation maps** $\mathcal{D}$. Given a variable $v \in \mathcal{V}$, $\mathcal{D}(v)$ denotes a disambiguation of $v$ for $G$. Given an operator $o \in \mathcal{O}$ and $v \in \mathcal{V}(\mathrm{eff}(o))$, $\mathcal{D}(o, v)$ denotes a disambiguation of $v$ for $\mathrm{pre}(o)$.

## 4 Potential Heuristics

Potential heuristics (Pommerening et al. 2015) were introduced as admissible and consistent heuristics that assign a numerical value (potential) to each fact, and the h-value for a state $s$ is a sum of the potentials of all facts in $s$. It was shown that potentials can be computed by solving a linear program (LP) with constraints expressing goal-awareness and consistency of the resulting heuristic. Pommerening, Helmert, and Bonet (2017) showed that this concept can be extended to larger sets of facts (higher-dimensional features/conjunctions) so that a potential is associated with each conjunction, and the h-value for a state $s$ is a sum over the potentials corresponding to conjunctions that are true in $s$.

Increasing the size of conjunctions allows to distinguish between more states, eventually leading to a potential heuristic that is optimal. However, the computation of potentials becomes significantly more difficult as the size of conjunctions grows. Pommerening, Helmert, and Bonet (2017) showed that deciding consistency of a potential heuristic is

coNP-hard in general if we consider all conjunctions of size 3 or more. They were however able to identify conditions when the construction of an admissible potential heuristic is tractable, depending on the interactions between the conjunctions. Steinmetz and Hoffmann (2018) showed that admissible higher-dimensional potential heuristics can be computed via a detour to a single-fact potential heuristics in the $\Pi^{\mathcal{C}}$ compilation, which we build upon here. The size of $\Pi^{\mathcal{C}}$ grows worst-case exponentially in $|\mathcal{C}|$, yielding an alternative tractability condition—when $\Pi^{\mathcal{C}}$ does not explode.

Since we plan to compute potential heuristics via compilations where each conjunction is explicitly represented as a fact, we formally use only a single-fact potential heuristics.

**Definition 3.** A **potential function** is a function $\mathsf{P} : \mathcal{F} \mapsto \mathbb{R}$. A **potential heuristic** for $\mathsf{P}$ maps each state $s \in \mathcal{R}(\Pi)$ to the sum of potentials of facts in $s$, i.e., $h^{\mathsf{P}}(s) = \sum_{f \in s} \mathsf{P}(f)$.

Potential functions inducing admissible potential heuristics can be found by solving LPs, and it was shown by Fišer, Horčík, and Komenda (2020) that, if restricted to the reachable states, potential heuristics can be strengthened by taking disambiguations into account. So, given a disambiguation map $\mathcal{D}$, we can find potential functions $\mathsf{P}$ by solving the following LP: The LP has a variable $\mathsf{P}(f)$ for each fact $f \in \mathcal{F}$, the constraint

$$\sum_{V \in \mathcal{V}} \max_{f \in \mathcal{D}(V)} \mathsf{P}(f) \leq 0$$

ensuring goal-awareness, and the constraint

$$\sum_{V \in \mathcal{V}(\mathrm{eff}(o))} \max_{f \in \mathcal{D}(o,V)} \mathsf{P}(f) - \sum_{f \in \mathrm{eff}(o)} \mathsf{P}(f) \leq \mathrm{cost}(o)$$

for each operator $o \in \mathcal{O}$ ensuring consistency. Note that the maximization can be easily implemented with auxiliary variables as described by Pommerening et al. (2015). Since the aforementioned constraints ensure goal-awareness and consistency (and therefore admissibility), the objective function of the LP can be freely chosen (Seipp, Pommerening, and Helmert 2015; Fišer, Horčík, and Komenda 2020), e.g., maximization of $\sum_{f \in I} \mathsf{P}(f)$ will result in a potential heuristic with the maximum possible h-value for the initial state.

# 5 $\Pi^{\mathcal{C}}$ Compilation

The first compilation we use for computing potentials over conjunctions is the $\Pi^{\mathcal{C}}$ compilation introduced by Haslum (2012) in the context of strengthening delete-relaxation heuristics. Since then, it proved to be useful in different contexts too (e.g., Keyder, Hoffmann, and Haslum 2014; Fickert, Hoffmann, and Steinmetz 2016; Steinmetz and Hoffmann 2018). In particular, Steinmetz and Hoffmann (2018) already showed that potential heuristics over $\Pi^{\mathcal{C}}$ provide consistent and admissible estimates for the original task. We follow up on their work in that we use the $\Pi^{\mathcal{C}}$ compilation in the FDR formalism (Haslum (2012) used STRIPS), but we also fully utilize mutexes to prune unreachable operators, as it was already pointed out by Keyder, Hoffmann, and Haslum (2014) that mutexes are effective in preventing the compilation to blow-up in practice as the size of $\Pi^{\mathcal{C}}$ is

worst-case exponential in $|\mathcal{C}|$. Another subtle difference to the work of Steinmetz and Hoffmann is that we consider heuristics defined over reachable states only. For these reasons, we provide full proofs showing that the compilation preserves admissibility and consistency of heuristics (moved to the technical report for space reasons). Moreover, we show that $\Pi^{\mathcal{C}}$ has some disadvantages. Namely, it can induce superfluous paths in the state space. For the rest of this section, let $\Pi = \langle \mathcal{V}, \mathcal{O}, I, G \rangle$ denote a task with facts $\mathcal{F}$, let $\mathcal{M}$ be a mutex-set, and let $\mathcal{C} \subseteq 2^{\mathcal{F}}$ be a set of conjunctions consisting of at least two facts (i.e., $|c| \geq 2$ for every $c \in \mathcal{C}$).

The idea of the $\Pi^{\mathcal{C}}$ compilation is following. First, we encode each conjunction $c \in \mathcal{C}$ as a binary variable $v_c$: If $c$ is true in a state $s$ ($c \subseteq s$), $v_c$ is set to 1, otherwise it is set to 0. Second, we set the initial state and goal accordingly. Finally, we construct multiple operators for each input operator $o \in \mathcal{O}$ so that application of at least one of them maintain the intended values of the $v_c$ variables—this is where the worst-case exponential blow-up comes from as we need to enumerate possible contexts in which each operator can be applied. Now, we introduce some auxiliary notation that will be helpful in the construction of operators.

Clearly, $o \in \mathcal{O}$ can make $c \in \mathcal{C}$ true or false only if $\mathrm{eff}(o)$ and $c$ share some variables: $\mathcal{C}^{\mathrm{eff}}(o) = \{c \in \mathcal{C} \mid \mathcal{V}(c) \cap \mathcal{V}(\mathrm{eff}(o)) \neq \emptyset\}$. Nevertheless, we are interested only in conjunctions whose truth values are actually changed by the operator $o$. Namely, conjunctions $c \in \mathcal{C}^{\mathrm{eff}}(o)$ that are either true before $o$ is applied and false after, or false before and true after. For every $c \in \mathcal{C}$ that is true before applying $o$ it holds that $c \cup \mathrm{pre}(o) \notin \mathcal{M}$ (as the opposite implies $c$ is false). Moreover, $o$ can make $c$ false only if $c \cup \mathrm{app}(o) \in \mathcal{M}$ (as $o$ must affect $c$ by changing at least one variable of $c$ to a different value). Unfortunately, we do not have any means to reliably test whether $c$ is false before the operator's application, but we at least know that $c$ can be made true by $o$ only if $c \cup \mathrm{app}(o) \notin \mathcal{M}$. This leads to the set of conjunctions possibly affected by $o \in \mathcal{O}$ (reduced using mutexes):

$$\mathcal{C}^{\mathrm{a}}(o) = \{c \in \mathcal{C}^{\mathrm{eff}}(o) \mid c \cup \mathrm{pre}(o) \notin \mathcal{M}, c \cup \mathrm{app}(o) \in \mathcal{M}\}$$
$$\cup \{c \in \mathcal{C}^{\mathrm{eff}}(o) \mid c \cup \mathrm{app}(o) \notin \mathcal{M}\}.$$

$\mathcal{C}^{\mathrm{a}}(o)$ can be partitioned into the set of conjunctions always made true by $o$ no matter what was their truth values before applying $o$:

$$\mathcal{C}^{\mathrm{t}}(o) = \{c \in \mathcal{C}^{\mathrm{a}}(o) \mid c \subseteq \mathrm{app}(o)\},$$

the set of conjunctions always made false by $o$:

$$\mathcal{C}^{\mathrm{f}}(o) = \{c \in \mathcal{C}^{\mathrm{a}}(o) \mid c \cup \mathrm{app}(o) \in \mathcal{M}\},$$

and the set of conjunctions potentially made true by $o$ depending on the state where $o$ is applied:

$$\mathcal{C}^{\mathrm{p}}(o) = \{c \in \mathcal{C}^{\mathrm{a}}(o) \mid c \cup \mathrm{app}(o) \notin \mathcal{M}, c \nsubseteq \mathrm{app}(o)\}.$$

Lastly, for every subset $X \subseteq \mathcal{C}$, we define the extension of the precondition of $o$ by parts of conjunctions from $X$ unaffected by $o$ (i.e., facts from $X$ that if true before the application, are true also after the operator's application):

$$\mathrm{ext\text{-}pre}(o, X) = \mathrm{pre}(o) \cup \bigcup_{c \in X} (c \setminus \mathrm{eff}(o)).$$

Now we are ready to formally define the $\Pi^{\mathcal{C}}$ compilation.

**Definition 4.** Given a planning task $\Pi$, a mutex-set $\mathcal{M}$, and a set of conjunctions $\mathcal{C} \subseteq 2^{\mathcal{F}}$ s.t. $|c| \geq 2$ for every $c \in \mathcal{C}$, the planning task $\Pi^{\mathcal{C}} = \langle \mathcal{V}^{\mathcal{C}}, \mathcal{O}^{\mathcal{C}}, I^{\mathcal{C}}, G^{\mathcal{C}} \rangle$ is defined as follows.

(1) $\mathcal{V}^{\mathcal{C}}$ extends $\mathcal{V}$ with a fresh binary variable $v_c$ for each conjunction $c \in \mathcal{C}$, i.e., $\mathcal{V}^{\mathcal{C}} = \mathcal{V} \cup \{v_c \mid c \in \mathcal{C}\}$, and, for each $c \in \mathcal{C}$, $v_c \notin \mathcal{V}$, $\mathrm{dom}(v_c) = \{0, 1\}$.

(2) The initial state is extended with the correct truth values of $v_c$ variables, i.e.,

$$I^{\mathcal{C}} = I \cup \{\langle v_c, 1 \rangle \mid c \in \mathcal{C}, c \subseteq I\} \cup \{\langle v_c, 0 \rangle \mid c \in \mathcal{C}, c \not\subseteq I\}.$$

(3) The goal is extended with $v_c$ set to 1 whenever $c$ is true in $G$, i.e., $G^{\mathcal{C}} = G \cup \{\langle v_c, 1 \rangle \mid c \in \mathcal{C}, c \subseteq G\}$.

(4) For every operator $o \in \mathcal{O}$ and every subset of conjunctions $X \subseteq \mathcal{C}^p(o)$ potentially made true by $o$ such that (i) ext-pre$(o, X) \notin \mathcal{M}$, and (ii) $X$ is downward closed on $\mathcal{C}^p(o)$ (i.e., for every $c \in \mathcal{C}^p(o)$ such that there exists $c' \in X$ such that $c \subseteq c'$ it holds that $c \in X$), $\mathcal{O}^{\mathcal{C}}$ has the operator $o^X$ with $\mathrm{cost}(o^X) = \mathrm{cost}(o)$,

$$\begin{aligned}
\mathrm{pre}(o^X) = {} & \text{ext-pre}(o, X) \\
& \cup \{\langle v_c, 1 \rangle \mid c \in \mathcal{C}, c \subseteq \text{ext-pre}(o, X)\}, \\
\mathrm{eff}(o^X) = {} & (\mathrm{eff}(o) \cup \{\langle v_c, 1 \rangle \mid c \in \mathcal{C}^t(o) \cup X\} \\
& \cup \{\langle v_c, 0 \rangle \mid c \in \mathcal{C}^f(o)\}) \setminus \mathrm{pre}(o^X).
\end{aligned}$$

$\Pi^{\mathcal{C}}$ is well-defined as all variables have finite domains, $I^{\mathcal{C}}$ is defined over all variables, and $\mathrm{pre}(o^X) \cap \mathrm{eff}(o^X) = \emptyset$ for all $o^X \in \mathcal{O}^{\mathcal{C}}$. Note that the blow-up of the compilation comes from enumerating all possible combinations of conjunctions that are potentially made true by an operator (4). It is, however, mitigated by skipping operators that can be proved to be unreachable (4i), and by considering only downward closed sets (4ii). The later comes from a simple observation that having $c, c' \in \mathcal{C}$ such that $c' \subseteq c$ and $c$ is true, then $c'$ is also necessarily true and therefore there is no reason for splitting the context where $c$ is true, $c'$ is true, and both are true.

Given a state $s$ from $\Pi$, we define the shorthand

$$\mathcal{C}[s] = s \cup \{\langle v_c, 1 \rangle \mid c \in \mathcal{C}, c \subseteq s\} \cup \{\langle v_c, 0 \rangle \mid c \in \mathcal{C}, c \not\subseteq s\}$$

that can be used for translating a state $s$ from $\Pi$ to a state in $\Pi^{\mathcal{C}}$ where all variables $v_c$ have assigned the correct values. Every path in $\Pi^{\mathcal{C}}$ corresponds to a path in $\Pi$, i.e., $\Pi^{\mathcal{C}}$ preserves applicability of the original operators from $\Pi$:

**Proposition 5.** Let $\pi = \langle o_1^{X_1}, \ldots, o_n^{X_n} \rangle$ denote a path in $\Pi^{\mathcal{C}}$. Then $\pi' = \langle o_1, \ldots, o_n \rangle$ is a path in $\Pi$ and $\pi'[\![I]\!] = \pi[\![I^{\mathcal{C}}]\!] \cap \mathcal{F}$.

Every path $\pi$ in $\Pi$ has its corresponding path $\pi'$ in $\Pi^{\mathcal{C}}$ with exactly the same length, cost, and it leads to a state in $\Pi^{\mathcal{C}}$ with correctly set variables $v_c$, i.e., $\pi'[\![I^{\mathcal{C}}]\!] = \mathcal{C}[\pi[\![I]\!]]$.

**Proposition 6.** Let $\pi = \langle o_1, \ldots, o_n \rangle$ denote a path in $\Pi$. Then there exists a path $\pi' = \langle o_1^{X_1}, \ldots, o_n^{X_n} \rangle$ in $\Pi^{\mathcal{C}}$ such that $\pi'[\![I^{\mathcal{C}}]\!] = \mathcal{C}[\pi[\![I]\!]]$.

Propositions 5 and 6 are enough to show that $\Pi^{\mathcal{C}}$ preserves consistency and admissibility of heuristics. To be precise, we show that if we have an admissible (consistent) heuristic $h^{\mathcal{C}}$ for $\Pi^{\mathcal{C}}$, then we can cast any state $s$ reachable in $\Pi$ to another state $\mathcal{C}[s]$ in $\Pi^{\mathcal{C}}$ and use $h^{\mathcal{C}}(\mathcal{C}[s])$ as an admissible (consistent) estimate for $s$ in $\Pi$.

**Theorem 7.** Let $h^{\mathcal{C}} : \mathcal{R}(\Pi^{\mathcal{C}}) \mapsto \mathbb{R}_0^+$ be a heuristic for $\Pi^{\mathcal{C}}$, let $h : \mathcal{R}(\Pi) \mapsto \mathbb{R}_0^+$ be a function s.t. $h(s) = h^{\mathcal{C}}(\mathcal{C}[s])$ for every $s \in \mathcal{R}(\Pi)$. If $h^{\mathcal{C}}$ is admissible (consistent) heuristic for $\Pi^{\mathcal{C}}$, then $h$ is an admissible (consistent) heuristic for $\Pi$.

*Proof.* From Proposition 6 it follows that $h$ is well-defined, because $\mathcal{C}[s] \in \mathcal{R}(\Pi^{\mathcal{C}})$ for every $s \in \mathcal{R}(\Pi)$. Every plan in $\Pi^{\mathcal{C}}$ has its corresponding plan in $\Pi$ with the same cost (Proposition 5), and every plan in $\Pi$ has its corresponding plan in $\Pi^{\mathcal{C}}$ with the same cost (Proposition 6). So, admissibility and consistency follows. $\square$

$\Pi^{\mathcal{C}}$ also has some pitfalls. Consider a task with a plan $\pi = \langle o_1, \ldots, o_n \rangle$ and $\mathcal{C} = \{c\}$ such that $c \not\subseteq I$ and $c \subseteq G$ and $c \not\subseteq \mathrm{eff}(o_i)$ for every $i \in \{1, \ldots, n\}$. From the construction of $\Pi^{\mathcal{C}}$ it follows that for every $o_i \in \mathcal{O}$ there is $o_i^{\emptyset} \in \mathcal{O}^{\mathcal{C}}$ constructed from the empty subset of $\mathcal{C}^p(o_i)$, and the sequence $\pi' = \langle o_1^{\emptyset}, \ldots, o_n^{\emptyset} \rangle$ is applicable in $I^{\mathcal{C}}$. Therefore, the resulting state $s = \pi'[\![I^{\mathcal{C}}]\!]$ contains all goal facts from $\Pi$, i.e., $G \subseteq s$, but $v_c$ is set to zero in $s$, i.e., $\langle v_c, 0 \rangle \in s$, even though $c \subseteq G \subseteq s$. In other words, we have that $s = \pi'[\![I^{\mathcal{C}}]\!] \neq \mathcal{C}[\pi[\![I]\!]]$. This does not contradict the propositions above. There still is another sequence of operators $\pi'' \neq \pi'$ from $\Pi^{\mathcal{C}}$ s.t. $\pi''[\![I^{\mathcal{C}}]\!] = \mathcal{C}[\pi[\![s]\!]]$ (Proposition 6), and it, indeed, holds that $\pi'[\![I^{\mathcal{C}}]\!] \cap \mathcal{F} = \pi[\![I]\!]$ (Proposition 5). However, it shows that the construction of $\Pi^{\mathcal{C}}$ can induce superfluous paths in the state space. These paths cannot be shortcuts, but they can be detours or lead to dead-ends.

## 6 $\Pi_{\mathrm{exact}}^{\mathcal{C}}$ Compilation

Here, we introduce a new compilation $\Pi_{\mathrm{exact}}^{\mathcal{C}}$ that, in contrast to $\Pi^{\mathcal{C}}$, preserves the reachable part of the state space exactly. $\Pi_{\mathrm{exact}}^{\mathcal{C}}$ follows $\Pi^{\mathcal{C}}$ in that it also encodes each $c \in \mathcal{C}$ as a binary variable $v_c$, but it differs from $\Pi^{\mathcal{C}}$ in the way operators are encoded. Each operator $o$ in $\Pi_{\mathrm{exact}}^{\mathcal{C}}$ explicitly encodes truth values of all conjunctions potentially affected by $o$ in both precondition and effect. This can lead to even larger blow-up than in $\Pi^{\mathcal{C}}$, but allows us to prove that not only all paths from $\Pi$ are preserved in $\Pi_{\mathrm{exact}}^{\mathcal{C}}$, but also that every reachable state $s$ in $\Pi_{\mathrm{exact}}^{\mathcal{C}}$ is of a form $s = \mathcal{C}[s \cap \mathcal{F}]$, i.e., in every reachable state, every $v_c$ is set to 1 whenever $c \subseteq s$, and it is set to 0 otherwise. Therefore, the reachable parts of state spaces of $\Pi$ and $\Pi_{\mathrm{exact}}^{\mathcal{C}}$ are isomorphic: There is one-to-one mapping between reachable states in $\Pi$ and $\Pi_{\mathrm{exact}}^{\mathcal{C}}$, one-to-one mapping between paths preserving costs, and therefore also one-to-one mapping between heuristics. From now on, let $\Pi = \langle \mathcal{V}, \mathcal{O}, I, G \rangle$, $\mathcal{F}$, $\mathcal{M}$, and $\mathcal{C} \subseteq 2^{\mathcal{C}}$ be as in the previous section. We use some auxiliary notation:

We use the same set $\mathcal{C}^a(o) \subseteq \mathcal{C}$ of conjunctions possibly affected by $o \in \mathcal{O}$. Moreover, we use total functions $f : \mathcal{C}^a(o) \mapsto \{0, 1\}$ for generating possible assignments to variables $v_c$ for all $c \in \mathcal{C}^a(o)$. So, given $o \in \mathcal{O}$ and a total function $f : \mathcal{C}^a(o) \mapsto \{0, 1\}$, we define sets of conjunctions affected by $o$ mapped by $f$ to 1 and 0, respectively:

$$\mathcal{C}^{f,1}(o) = \{c \in \mathcal{C}^a(o) \mid f(c) = 1\},$$

$$\mathcal{C}^{f,0}(o) = \{c \in \mathcal{C}^a(o) \mid f(c) = 0\},$$

and the set of variables from $\mathcal{C}^{f,0}(o)$ that are not part of the precondition or effect of $o$:

$$\mathcal{V}^{f,0}(o) = \bigcup_{c \in \mathcal{C}^{f,0}(o)} \mathcal{V}(c) \setminus \mathcal{V}(\mathrm{pre}(o) \cup \mathrm{eff}(o)).$$

For every partial state $p$ s.t. $\mathcal{V}(p) = \mathcal{V}^{f,0}(o)$, we define

$$\mathrm{expre}(o, f, p) = \mathrm{pre}(o) \cup p \cup \bigcup_{c \in \mathcal{C}^{f,1}(o)} c,$$

$$\mathrm{exapp}(o, f, p) = (\mathrm{expre}(o, f, p) \setminus \mathcal{F}_{\mathrm{eff}(o)}) \cup \mathrm{eff}(o).$$

The idea of the construction of operators is that for every operator $o \in \mathcal{O}$, we consider all possible assignments to variables $v_c$ corresponding to possibly affected conjunctions $c \in \mathcal{C}^a(o)$. For each such assignment (a total function $f : \mathcal{C}^a(o) \mapsto \{0,1\}$), we create a set of operators based on $o$ applicable only in states where the assignment holds. Each of these operators correspond to a possible change of the $v_c$ variables. An operator $o$ changes the value of $v_c$ from 1 to 0 whenever $\mathrm{eff}(o)$ has at least one common variable with $c$ but the values differ. Changing the value of $v_c$ from 0 to 1 is more complicated. Clearly, $o$ can change $v_c$ to 1 only if $\mathrm{eff}(o)$ and $c$ agree on all common variables, but we also need to consider an additional context of assignments to variables from $c$ that are not in $\mathrm{pre}(o)$ or $\mathrm{eff}(o)$ (i.e., variables $\mathcal{V}^{f,0}(o) \cap \mathcal{V}(c)$). Consider a conjunction $c = \{\langle a, 0 \rangle, \langle b, 0 \rangle\}$ and an operator $o$ with $\mathrm{pre}(o) = \{\langle a, 1 \rangle\}$ and $\mathrm{eff}(o) = \{\langle a, 0 \rangle\}$. If $c$ is false in some state $s$, then either $\langle a, 0 \rangle \notin s$ or $\langle b, 0 \rangle \notin s$. So clearly, $o$ can make $c$ true only if $\langle b, 0 \rangle \in s$, as the effect assigns 0 to the variable $a$, but the value of $b$ is not specified in the precondition of the operator. In other words, $o$ makes $c$ true depending on the additional context which is not explicitly specified in its precondition. Therefore, the construction iterates over all possible contexts of variables $\mathcal{V}^{f,0}(o)$ so that we are able to exactly determine when the variables $v_c$ change their values from 0 to 1.

So, the blow-up of the $\Pi^{\mathcal{C}}_{\mathrm{exact}}$ compilation is worst-case exponential in $|\mathcal{C}|$ and $|\mathcal{V}|$, but we can mitigate it by using mutexes as we need to consider only contexts that can co-occur with operator's precondition.

**Definition 8.** Given a planning task $\Pi$, a mutex-set $\mathcal{M}$, and a set of conjunctions $\mathcal{C} \subseteq 2^{\mathcal{F}}$ s.t. $|c| \geq 2$ for every $c \in \mathcal{C}$, the planning task $\Pi^{\mathcal{C}}_{\mathrm{exact}} = \langle \mathcal{V}^{\mathcal{C}}, \mathcal{O}^{\mathcal{C}}_{\mathrm{exact}}, I^{\mathcal{C}}, G^{\mathcal{C}} \rangle$ is defined as follows. $\mathcal{V}^{\mathcal{C}}$, $I^{\mathcal{C}}$, and $G^{\mathcal{C}}$ are exactly the same as for $\Pi^{\mathcal{C}}$, but $\mathcal{O}^{\mathcal{C}}_{\mathrm{exact}}$ is constructed differently: For every operator $o \in \mathcal{O}$ and every total function $f : \mathcal{C}^a(o) \mapsto \{0,1\}$ and every partial state $p$ over variables $\mathcal{V}^{f,0}(o)$ such that (i) $\mathrm{expre}(o, f, p) \notin \mathcal{M}$ and (ii) for every $c \in \mathcal{C}^{f,0}(o)$ it holds that $c \not\subseteq \mathrm{expre}(o, f, p)$, $\mathcal{O}^{\mathcal{C}}_{\mathrm{exact}}$ has the operator $o^{f,p}$ such that $\mathrm{cost}(o^{f,p}) = \mathrm{cost}(o)$,

$$\mathrm{pre}(o^{f,p}) = \mathrm{expre}(o, f, p) \cup \{\langle v_c, f(c) \rangle \mid c \in \mathcal{C}^a(o)\},$$

$$\begin{aligned} \mathrm{eff}(o^{f,p}) = \; & \mathrm{eff}(o) \setminus \mathrm{pre}(o^{f,p}) \\ & \cup \{\langle v_c, 0 \rangle \mid c \in \mathcal{C}^{f,1}(o), c \cup \mathrm{app}(o) \in \mathcal{M}\} \\ & \cup \{\langle v_c, 1 \rangle \mid c \in \mathcal{C}^{f,0}(o), c \subseteq \mathrm{exapp}(o, f, p)\}. \end{aligned}$$

As for $\Pi^{\mathcal{C}}$, it is easy to see that $\Pi^{\mathcal{C}}_{\mathrm{exact}}$ is well-defined. Note that the condition (ii) makes sure that we consider

only compatible (downward closed) assignments to variables, i.e., whenever we have two conjunctions $c, c' \in \mathcal{C}$ such that $c' \subseteq c$ and $f(v_c) = 1$, then the compilation considers only the functions $f$ where also $f(v_{c'}) = 1$. Also note that effects set $v_c$ to 1 only if $v_c$ is set to 0 in the precondition and vice versa $\langle v_c, 0 \rangle \in \mathrm{eff}(o^{f,p})$ only if $\langle v_c, 1 \rangle \in \mathrm{pre}(o^{f,p})$.

Analogously to the previous section, we use $\mathcal{C}[s]$ to translate states from $\Pi$ to $\Pi^{\mathcal{C}}_{\mathrm{exact}}$. Now, we show that every path in $\Pi^{\mathcal{C}}_{\mathrm{exact}}$ has its counterpart with the same cost in $\Pi$.

**Proposition 9.** *Let* $\pi = \langle o_1^{f_1, p_1}, \ldots, o_n^{f_n, p_n} \rangle$ *denote a path in* $\Pi^{\mathcal{C}}_{\mathrm{exact}}$. *Then* $\pi' = \langle o_1, \ldots, o_n \rangle$ *is a path in* $\Pi$ *and* $\pi'\llbracket I \rrbracket = \pi\llbracket I^{\mathcal{C}} \rrbracket \cap \mathcal{F}$.

*Proof.* Let $s_0, s_1, \ldots, s_n$ denote the intermediate states of $\pi$ applied on the initial state of $\Pi^{\mathcal{C}}_{\mathrm{exact}}$, i.e., $s_0 = I^{\mathcal{C}}$ and for every $i \in \{1, \ldots, n\}$ it holds that $o_i^{f_i, p_i}\llbracket s_{i-1} \rrbracket = s_i$. Moreover, let $s_i' = s_i \cap \mathcal{F}$ for every $i \in \{0, 1, \ldots, n\}$. Now, we show that $s_0' = I$ and, for every $i \in \{1, \ldots, n\}$, $s_i'$ is a state in $\Pi$ reachable by the sequence of operators $\langle o_1, \ldots, o_i \rangle$, and $s_i' = s_i \cap \mathcal{F}$.

From the definition of $\Pi^{\mathcal{C}}_{\mathrm{exact}}$ we have that $I^{\mathcal{C}} \cap \mathcal{F} = I = s_0'$. Since, for every $i \in \{1, \ldots, n\}$, it holds that $\mathrm{pre}(o_i) \subseteq \mathrm{expre}(o_i, f_i, p_i) \subseteq \mathrm{pre}(o_i^{f_i, p_i}) \subseteq s_{i-1}$, it follows that $\mathrm{pre}(o_i) \subseteq s_{i-1}'$, i.e., $o_i$ is applicable in $s_{i-1}'$. Since $\mathrm{eff}(o_i^{f_i, p_i}) \cap \mathcal{F} = \mathrm{eff}(o) \setminus \mathrm{pre}(o_i^{f_i, p_i})$ it follows that for every $f \in \mathrm{eff}(o)$ it holds that either $f \in \mathrm{pre}(o_i^{f_i, p_i})$ (and therefore also $f \in s_{i-1}'$ and $f \in s_i'$) or $f \in \mathrm{eff}(o_i^{f_i, p_i})$ (and therefore $f \in s_i'$). Finally, since $\mathrm{eff}(o_i^{f_i, p_i}) \cap \mathcal{F} \subseteq \mathrm{eff}(o)$ we have $o_i^{f_i, p_i}\llbracket s_{i-1} \rrbracket \cap \mathcal{F} = o_i\llbracket s_{i-1}' \rrbracket = s_i'$. $\square$

Next, we show that for every reachable state $s$ in $\Pi^{\mathcal{C}}_{\mathrm{exact}}$ it holds that $v_c$ is set to 1 whenever $c$ is true in $s$, and $v_c$ is set to 0 whenever $c$ is false in $s$. In other words, in contrast to $\Pi^{\mathcal{C}}$, all paths in $\Pi^{\mathcal{C}}$ lead to states with correctly set $v_c$ values.

**Proposition 10.** *Let* $s \in \mathcal{R}(\Pi^{\mathcal{C}}_{\mathrm{exact}})$ *denote a reachable state in* $\Pi^{\mathcal{C}}_{\mathrm{exact}}$. *Then* $s = \mathcal{C}[s \cap \mathcal{F}]$.

*Proof.* Since $I^{\mathcal{C}} = \mathcal{C}[I^{\mathcal{C}} \cap \mathcal{F}]$ by construction, it follows that if there exists a reachable state $s \in \mathcal{R}(\Pi^{\mathcal{C}}_{\mathrm{exact}})$ such that $s \neq \mathcal{C}[s \cap \mathcal{F}]$, then there also exist a reachable state $s' \in \mathcal{R}(\Pi^{\mathcal{C}}_{\mathrm{exact}})$ and an operator $o^{f,p} \in \mathcal{O}^{\mathcal{C}}_{\mathrm{exact}}$ applicable in $s'$ such that $s' = \mathcal{C}[s' \cap \mathcal{F}]$ and $o^{f,p}\llbracket s' \rrbracket = s$.

We need to investigate four cases:

(i) There exists $c \in \mathcal{C}$ such that $c \subseteq s'$ and $\langle v_c, 1 \rangle \in s'$ and $c \subseteq s$ and $\langle v_c, 0 \rangle \in s$: The assignment to $v_c$ can change to $\langle v_c, 0 \rangle$ only if $c \cup \mathrm{app}(o) \in \mathcal{M}$ which is in contradiction with $c \subseteq s$ because $\mathrm{app}(o) \subseteq s$.

(ii) There exists $c \in \mathcal{C}$ such that $c \subseteq s'$ and $\langle v_c, 1 \rangle \in s'$ and $c \not\subseteq s$ and $\langle v_c, 1 \rangle \in s$: From $c \subseteq s'$ and $c \not\subseteq s$ we have that $c \cup \mathrm{app}(o) \in \mathcal{M}$, therefore $\langle v_c, 0 \rangle \in s$ by construction of $\mathrm{eff}(o^{f,p})$, which is in contradiction with $\langle v_c, 1 \rangle \in s$.

(iii) There exists $c \in \mathcal{C}$ such that $c \not\subseteq s'$ and $\langle v_c, 0 \rangle \in s'$ and $c \not\subseteq s$ and $\langle v_c, 1 \rangle \in s$: The assignment to $v_c$ can change to $\langle v_c, 1 \rangle$ only if $c \subseteq \mathrm{exapp}(o, f, p)$ which is in contradiction with $c \not\subseteq s$ because $\mathrm{exapp}(o, f, p) \subseteq s$.

(iv) There exists $c \in \mathcal{C}$ such that $c \not\subseteq s'$ and $\langle v_c, 0 \rangle \in s'$ and $c \subseteq s$ and $\langle v_c, 0 \rangle \in s$: From $c \not\subseteq s'$ and $c \subseteq s$ and

Proposition 9 it follows that $c \setminus s' \subseteq \mathrm{eff}(o)$, and from the construction $\mathrm{expre}(o, f, p)$ we have that $\mathcal{V}(c \setminus \mathrm{eff}(o)) \subseteq \mathcal{V}(\mathrm{expre}(o, f, p))$, therefore $\mathcal{V}(c) \subseteq \mathcal{V}(\mathrm{exapp}(o, f, p))$, therefore $c \subseteq \mathrm{exapp}(o, f, p)$ because $\mathrm{exapp}(o, f, p) \subseteq s$, therefore $\langle v_c, 1 \rangle \in s$ which contradicts $\langle v_c, 0 \rangle \in s$. $\square$

Next, we show that for every path in $\Pi$ there is a corresponding path in $\Pi_{\mathrm{exact}}^{\mathcal{C}}$ with exactly the same cost.

**Proposition 11.** *Let* $\pi = \langle o_1, \ldots, o_n \rangle$ *denote a path in* $\Pi$. *Then there exists a path* $\pi' = \langle o_1^{f_1, p_1}, \ldots, o_n^{f_n, p_n} \rangle$ *in* $\Pi_{\mathrm{exact}}^{\mathcal{C}}$ *and* $\pi'[\![I^{\mathcal{C}}]\!] = \mathcal{C}[\pi[\![I]\!]]$.

*Proof.* Let $s_0, s_1, \ldots, s_n$ denote the intermediate states of $\pi$ applied on $I$, i.e., $s_0 = I$ and for every $i \in \{1, \ldots, n\}$ it holds that $o_i[\![s_{i-1}]\!] = s_i$. Moreover, let $s_i' = \mathcal{C}[s_i]$ for every $i \in \{0, 1, \ldots, n\}$. From Definition 8 it follows that $s_0' = I^{\mathcal{C}}$.

Let assume that there exists a sequence of operators $\pi' = \langle o_1^{f_1, p_1}, \ldots, o_{i-1}^{f_{i-1}, p_{i-1}} \rangle$, for some $i < n$, such that $\pi'$ is applicable in $s_0'$ and $\pi'[\![s_0']\!] = s_{i-1}'$. Now, we show that there exists $o_i^{f_i, p_i}$ such that $o_i^{f_i, p_i}$ is applicable in $s_{i-1}'$ and $o_i^{f_i, p_i}[\![s_{i-1}']\!] = s_i'$. From Proposition 9 we have that $\mathrm{pre}(o_i) \subseteq s_{i-1} = s_{i-1}' \cap \mathcal{F}$. Let $f_i : \mathcal{C}^{\mathrm{a}}(o) \mapsto \{0, 1\}$ denote a total function such that for every $c \in \mathcal{C}^{\mathrm{a}}(o)$ it holds that $\langle v_c, f(c) \rangle \in s_{i-1}'$, and let $p_i = s_{i-1}' \cap \mathcal{F}_{\mathcal{V}^{f_i, 0}(o_i)}$. It is easy to see that $f_i$ is well-defined because $\mathcal{V}(s_{i-1}') = \mathcal{V}$. It is also easy to see that $p_i$ is a partial state, and since we have that $\mathrm{pre}(o_i) \subseteq s_{i-1}'$ and $p \subseteq s_{i-1}'$ and, for every $c \in \mathcal{C}^{f_i, 1}(o_i)$, $c \subseteq s_{i-1}'$, it follows that $\mathrm{expre}(o_i) \notin \mathcal{M}$ and therefore $o_i^{f_i, p_i} \in \mathcal{O}_{\mathrm{exact}}^{\mathcal{C}}$, and for the same reasons we have that $\mathrm{pre}(o_i^{f_i, p_i}) \subseteq s_{i-1}'$. Therefore $o_i^{f_i, p_i}$ is applicable in $s_{i-1}'$ and the rest follows from Proposition 10. $\square$

Finally, we show that there is a one-to-one correspondence between heuristics in $\Pi$ and $\Pi_{\mathrm{exact}}^{\mathcal{C}}$ preserving admissibility and consistency. It is because every path in $\Pi$ can be mapped to a path in $\Pi_{\mathrm{exact}}^{\mathcal{C}}$ (Proposition 11), every path in $\Pi_{\mathrm{exact}}^{\mathcal{C}}$ can be mapped back to a path in $\Pi$ (Proposition 9), and there is a bijective mapping between all intermediate states of all paths in $\Pi$ and $\Pi_{\mathrm{exact}}^{\mathcal{C}}$ (Proposition 10).

**Theorem 12.** *(A) If* $h_{\mathrm{ex}}^{\mathcal{C}} : \mathcal{R}(\Pi_{\mathrm{exact}}^{\mathcal{C}}) \mapsto \mathbb{R}_0^+$ *is an admissible (consistent) heuristic for* $\Pi_{\mathrm{exact}}^{\mathcal{C}}$, *then* $h : \mathcal{R}(\Pi) \mapsto \mathbb{R}_0^+$ *such that* $h(s) = h_{\mathrm{ex}}^{\mathcal{C}}(\mathcal{C}[s])$ *for every* $s \in \mathcal{R}(\Pi)$ *is an admissible (consistent) heuristic for* $\Pi$.

*(B) If* $h : \mathcal{R}(\Pi) \mapsto \mathbb{R}_0^+$ *is an admissible (consistent) heuristic for* $\Pi$, *then* $h_{\mathrm{ex}}^{\mathcal{C}} : \mathcal{R}(\Pi_{\mathrm{exact}}^{\mathcal{C}}) \mapsto \mathbb{R}_0^+$ *such that* $h_{\mathrm{ex}}^{\mathcal{C}}(s) = h(s \cap \mathcal{F})$ *for every* $s \in \mathcal{R}(\Pi_{\mathrm{exact}}^{\mathcal{C}})$ *is admissible (consistent) heuristic for* $\Pi_{\mathrm{exact}}^{\mathcal{C}}$.

*Proof.* Let $\mathcal{T}(\Pi)$ denote the set of all paths in $\Pi$, and let $\mathcal{T}(\Pi_{\mathrm{exact}}^{\mathcal{C}})$ denote the set of all paths in $\Pi_{\mathrm{exact}}^{\mathcal{C}}$. From Propositions 9, 10, and 11, it follows there are bijective functions $f_t : \mathcal{T}(\Pi) \mapsto \mathcal{T}(\Pi_{\mathrm{exact}}^{\mathcal{C}})$ and $f_s : \mathcal{R}(\Pi) \mapsto \mathcal{R}(\Pi_{\mathrm{exact}}^{\mathcal{C}})$ such that $\mathrm{cost}(\pi) = \mathrm{cost}(f_t(\pi))$ for every $\pi \in \mathcal{T}(\Pi)$. $\square$

Despite the encouraging theoretical result formulated in Theorem 12, the construction of $\Pi_{\mathrm{exact}}^{\mathcal{C}}$ can result in having unreachable operators (as the input $\Pi$ can have unreachable operators too). Therefore, inferring potential heuristics

---

**Algorithm 1:** Inference of improving conjunctions.

**Input:** A task $\Pi$ with facts $\mathcal{F}$, an optimization criteria $\mathrm{Opt}$
**Output:** A set of conjunctions $\mathcal{C}_{\mathrm{o}}$

1   $\mathcal{C}_{\mathrm{o}} \leftarrow \emptyset$; $\mathrm{P} \leftarrow$ potential function for $\Pi$ maximizing $\mathrm{Opt}$;
2   **for each** $k = 2, \ldots, |\mathcal{V}|$ **do**
3     **for each** $c \subseteq \mathcal{F}, |c| = k, c \notin \mathcal{M}$ **do**
4       **if** *time limit reached* **then return** $\mathcal{C}_{\mathrm{o}}$ ;
5       $\mathcal{C} \leftarrow \mathcal{C}_{\mathrm{o}} \cup \{x \subseteq c \mid |x| \geq 2\}$;
6       $\mathrm{P}' \leftarrow$ potential func. for $\Pi^{\mathcal{C}}$ (or $\Pi_{\mathrm{exact}}^{\mathcal{C}}$) max. $\mathrm{Opt}$;
7       **if** $\mathrm{P}'$ *is improvement over* $\mathrm{P}$ **then**
8         $\mathcal{C}_{\mathrm{o}} \leftarrow \mathcal{C}$; $\mathrm{P} \leftarrow \mathrm{P}'$;
9         **go to** 2;
10 **return** $\mathcal{C}_{\mathrm{o}}$

---

as described in Section 4 does not guarantee we are able to find all possible potential heuristics (as it is not guaranteed in general even for $\Pi$).

## 7 Potential Heuristics over Conjunctions

Using potential heuristics over conjunctions in explicit-state search is straightforward. Given a set of conjunctions $\mathcal{C}$, we construct either $\Pi^{\mathcal{C}}$ or $\Pi_{\mathrm{exact}}^{\mathcal{C}}$ compilation of the input planning task $\Pi$, compute a potential heuristic $h^{\mathcal{C}}$ in $\Pi^{\mathcal{C}}$ or $\Pi_{\mathrm{exact}}^{\mathcal{C}}$ as described in Section 4, and finally use the heuristic $h$ such that $h(s) = h^{\mathcal{C}}(\mathcal{C}[s])$ for all reachable states $s \in \mathcal{R}(\Pi)$ when running search in $\Pi$. Theorems 7 and 12 show that such heuristic $h$ is admissible and consistent.

Moreover, $\Pi^{\mathcal{C}}$ and $\Pi_{\mathrm{exact}}^{\mathcal{C}}$ compilations are ordinary FDR tasks that preserve all optimal plans (and do not induce any shortcuts), therefore we can plan directly in them. Such approach does not offer many benefits for explicit-state search as the larger number of operators makes generating successor states slower, more variables lead to larger memory consumption for storing generated states, and the state space of $\Pi^{\mathcal{C}}$ is possibly larger than the state space of $\Pi$ (although this is not the case for $\Pi_{\mathrm{exact}}^{\mathcal{C}}$). However, there are planning techniques that might benefit from this approach.

Fišer, Torralba, and Hoffmann (2022a,b) showed that potential heuristics can be transformed into operator-potential heuristics associating each operator $o$ with the change of heuristic value of the corresponding potential heuristic induced by $o$. Consequently, operator-potential heuristics can be used to significantly improve performance of symbolic search that searches over sets of states (represented as binary decision diagrams (Bryant 1986)) rather than individual states. So, we can run this planning technique directly on $\Pi^{\mathcal{C}}$ and $\Pi_{\mathrm{exact}}^{\mathcal{C}}$ where the increased informativeness of potential heuristics automatically translates into operator-potential heuristics. This, of course, does not mean that the symbolic search will not suffer from using larger planning tasks. Also note that different variants of operators $o^X$ in $\Pi^{\mathcal{C}}$ ($o^{f,p}$ in $\Pi_{\mathrm{exact}}^{\mathcal{C}}$) corresponding to the same operator $o$ from $\Pi$ can induce a different change of heuristic values. Therefore, we cannot directly translate potential heuristics over conjunctions obtained via compilations to operator-potential heuristics in the original task $\Pi$.

Next question is how to obtain conjunctions $\mathcal{C}$ improv-

ing potential heuristics. As we already indicated before, we do not attempt do design an efficient way to do it here. Instead, we aim at gathering evidence that it is, indeed, common that there are small sets of conjunctions that can significantly improve potential heuristics. So, here we use a simple greedy algorithm (Algorithm 1) that systematically tries to test conjunctions one by one starting from the smallest ones (lines 2 and 3). For each tested conjunction and all its subsets (line 5), a compilation is constructed and the corresponding potential heuristic inferred using the LP described in Section 4 (line 6). Finally, if the potential heuristic is improvement over the current one, we extend the set of conjunctions (lines 7 and 8) and restart the whole process (line 9).

Potential heuristics can be computed by maximizing different optimization criteria—in our experiments we focus on the maximization of the h-value for the initial state only. The improvement of the resulting heuristics can also be measured in different ways. Here, we simply compare objective values of the corresponding LPs—since we maximize, higher values indicate a better heuristic. However, note that the inference using LP cannot guarantee dominance between potential heuristics in a general sense (one upper-bounds the other in *all* reachable states). It can only ensure dominance with respect to the optimization criteria (e.g., h-value for the initial state, or average h-value over all syntactic states).

## 8 Experimental Evaluation

The proposed method was implemented in C (source codes and data will be released) and evaluated on a cluster with Intel Xeon E5-2650v3 processors and 4 GB memory limit for each process. We used all planning domains from the optimal track of International Planning Competitions (IPCs) from 1998 to 2023 excluding the ones containing conditional effects after translation. We merged, for each domain, all benchmark from different IPCs leaving 54 domains and 1 806 tasks overall. Operators and facts are pruned with the $h^2$ heuristic in forward and backward direction (Alcázar and Torralba 2015), and the translation from PDDL to FDR uses the mutex groups inference proposed by Fišer (2020, 2023).

For explicit-state search, we ran $A^\star$ algorithm (Hart, Nilsson, and Raphael 1968) with two variants of potential heuristics: I denotes maximization of the h-value of the initial state (Pommerening et al. 2015), and A denotes maximization for the average (syntactic) state while enforcing the maximum h-value for the initial state (Seipp, Pommerening, and Helmert 2015; Fišer, Horčík, and Komenda 2020). We also evaluated symbolic search with operator-potential heuristics (Fišer, Torralba, and Hoffmann 2022a,b): We used forward search with A (denoted by $\overrightarrow{\text{A}}$), backward search with I ($\overleftarrow{\text{I}}$) and the combination of the two in the bidirectional search ($\overrightarrow{\text{A}}$-$\overleftarrow{\text{I}}$). We use 30 minutes time limit for all search variants (not counting the inference of conjunctions).

Since $A^\star$ with potential heuristics tend to terminate quickly, we also consider simple portfolios $P^\mathcal{C}$ and $P^\mathcal{C}_{\text{ex}}$: We run $A^\star$ with I or A in $\Pi$ until it terminates. Then, in the remainder of the 30 minutes time budget, we run Algorithm 1 for 5 minutes to infer conjunctions $\mathcal{C}$, and then run $A^\star$ again this time with potential heuristics over $\mathcal{C}$ using $\Pi^\mathcal{C}$ ($P^\mathcal{C}$) or

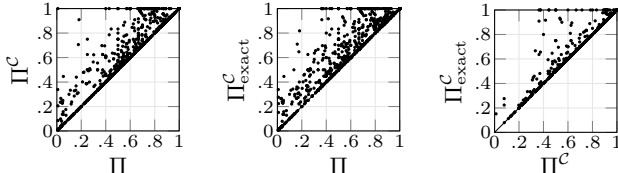

Figure 1: Comparison of h-values for initial states as ratios to the $h^\star$-values for tasks where $h^\star$-values are known. $\Pi^\mathcal{C}$ vs. $\Pi^\mathcal{C}_{\text{exact}}$ is compared over the same set of conjunctions inferred using $\Pi^\mathcal{C}_{\text{exact}}$.

$\Pi^\mathcal{C}_{\text{exact}}$ ($P^\mathcal{C}_{\text{ex}}$). These portfolios count the time spent in the inference of conjunctions into the whole time limit.

To obtain conjunctions $\mathcal{C}$, we ran Algorithm 1 for 5 minutes for each task, and we set the optimization criteria to the maximization of the h-value for the initial state. Improving conjunctions were found with $\Pi^\mathcal{C}$ ($\Pi^\mathcal{C}_{\text{exact}}$) in 47 (50) domains and 794 (844) tasks. So, $\Pi^\mathcal{C}_{\text{exact}}$ seems to be slightly more successful in this respect. The maximum size of the found conjunction per task ranged from 2 to 8 for $\Pi^\mathcal{C}$ and from 2 to 5 for $\Pi^\mathcal{C}_{\text{exact}}$. The average (median) was 2.5 (2) for $\Pi^\mathcal{C}$ and 2.6 (2) for $\Pi^\mathcal{C}_{\text{exact}}$ meaning most of the improvement comes from pairs of facts which is not surprising given Algorithm 1 tests all pairs first. The minimum, maximum, average and median size of the resulting $\mathcal{C}$ was, respectively, 1, 28, 3.9 and 3 for $\Pi^\mathcal{C}$, and 1, 38, 3.8 and 3 for $\Pi^\mathcal{C}_{\text{exact}}$. This indicates that both compilations used small sets of conjunctions in most cases. Fig. 1 compares h-values for initial states as ratios to the optimal heuristic on tasks where we knew $h^\star$ (1 333 tasks). These results show that it, indeed, is possible to obtain significantly more informative potential heuristics via $\Pi^\mathcal{C}$ and $\Pi^\mathcal{C}_{\text{exact}}$ with just few additional conjunctions.

To get some indication which compilation tends to lead to more informative heuristics, the comparison between $\Pi^\mathcal{C}$ and $\Pi^\mathcal{C}_{\text{exact}}$ in Fig. 1 is done with the same $\mathcal{C}$ (inferred with $\Pi^\mathcal{C}_{\text{exact}}$) in each task for both. It seems $\Pi^\mathcal{C}_{\text{exact}}$ is more successful than $\Pi^\mathcal{C}$ in this respect too. However, $\Pi^\mathcal{C}_{\text{exact}}$ also tends to generate larger task representations than $\Pi^\mathcal{C}$: The minimum, maximum, average and median ratio between the number of operators in the compilation and in the original planning task was 1, 7.9, 1.2 and 1 for $\Pi^\mathcal{C}$, and 1, 46.4, 2.8 and 1.4 for $\Pi^\mathcal{C}_{\text{exact}}$. This is also reflected in the speed of computing potential heuristics. We were able to evaluate on average 64.4 conjunctions per second in case of $\Pi^\mathcal{C}$ (median was 32.7), and 55.3 with $\Pi^\mathcal{C}_{\text{exact}}$ (median 27.6).

Table 1 shows the number of solved tasks (coverage) by the explicit-state search configurations. We are able to increase the coverage in some domains even with a small set of conjunctions inferred with a simple greedy uninformed algorithm. (Note that $A^\star$ with I over all pairs of facts solves only 451 tasks.) $\Pi^\mathcal{C}$ and $\Pi^\mathcal{C}_{\text{exact}}$ perform similarly, and the baseline ($\Pi$) rarely solves more tasks. The biggest difference between $\Pi^\mathcal{C}$ and $\Pi^\mathcal{C}_{\text{exact}}$ seems to be in A and the woodworking domain where $\Pi^\mathcal{C}$ managed to obtain more informed heuristics mainly due to the fact that Algorithm 1 with $\Pi^\mathcal{C}$ was able to test larger number of conjunctions within the time limit.

Portfolio results ($P^\mathcal{C}$, $P^\mathcal{C}_{\text{ex}}$) show that 30 minutes time bud-

| domain | A⋆ with I | | | | | A⋆ with A | | | | |
|---|---|---|---|---|---|---|---|---|---|---|
| | Π | Π^$\mathcal{C}$ | Π^$\mathcal{C}_{ex}$ | P^$\mathcal{C}$ | P^$\mathcal{C}_{ex}$ | Π | Π^$\mathcal{C}$ | Π^$\mathcal{C}_{ex}$ | P^$\mathcal{C}$ | P^$\mathcal{C}_{ex}$ |
| depot (22) | **6** | 5 | **6** | **6** | **6** | 11 | 11 | 11 | 11 | 11 |
| driverlog (20) | 8 | **9** | **9** | **9** | **9** | 13 | 13 | 13 | 13 | 13 |
| elevators (50) | 28 | **31** | **31** | **31** | **31** | 28 | **35** | 33 | **35** | 33 |
| freecell (80) | 48 | **49** | 47 | **49** | 48 | 70 | 70 | 70 | 70 | 70 |
| ged (20) | 15 | **19** | **19** | **19** | **19** | 15 | **19** | **19** | **19** | **19** |
| logistics (63) | 12 | **17** | **17** | **17** | **17** | 21 | **25** | **25** | **25** | **25** |
| nomystery (20) | 9 | **10** | 9 | **10** | 9 | 14 | 14 | 14 | 14 | 14 |
| openstacks (100) | **54** | 53 | 53 | **54** | **54** | **54** | 53 | 53 | **54** | **54** |
| parking (40) | 0 | **1** | **1** | **1** | **1** | 14 | 14 | **15** | 14 | **15** |
| pipesw-notank (50) | 24 | 24 | 24 | **25** | 24 | 26 | 26 | **27** | 26 | **27** |
| pipesw-tank (50) | 15 | **16** | 15 | **16** | 15 | 17 | **18** | 17 | **18** | 17 |
| rovers (40) | 6 | 6 | **7** | 6 | **7** | 8 | 8 | 8 | 8 | 8 |
| slitherlink (20) | **4** | **4** | 3 | **4** | **4** | 4 | 4 | 4 | 4 | 4 |
| spider (20) | **14** | 13 | **14** | **14** | **14** | 15 | 15 | 15 | 15 | 15 |
| tidybot (40) | **32** | **32** | 30 | **32** | **32** | 32 | 32 | 32 | 32 | 32 |
| tpp (30) | 6 | **8** | **8** | **8** | **8** | 8 | 8 | 8 | 8 | 8 |
| visitall (40) | 23 | 23 | **24** | 23 | **24** | 29 | 29 | 29 | 29 | 29 |
| woodworking (50) | 17 | **19** | **19** | **19** | **19** | 27 | **39** | 31 | **39** | 31 |
| others (1051) | 571 | 571 | 571 | 571 | 571 | 614 | 614 | 614 | 614 | 614 |
| Σ (1806) | 892 | 910 | 907 | **914** | 912 | 1020 | 1047 | 1038 | **1048** | 1039 |

Table 1: Coverage for explicit-state search.

| domain | Symb. $\overrightarrow{A}$ | | | Symb. $\overleftarrow{I}$ | | | Symb. $\overrightarrow{A}$ - $\overleftarrow{I}$ | | |
|---|---|---|---|---|---|---|---|---|---|
| | Π | Π^$\mathcal{C}$ | Π^$\mathcal{C}_{ex}$ | Π | Π^$\mathcal{C}$ | Π^$\mathcal{C}_{ex}$ | Π | Π^$\mathcal{C}$ | Π^$\mathcal{C}_{ex}$ |
| agricola (20) | **19** | 16 | 18 | 4 | 0 | 1 | **18** | 12 | 16 |
| barman (34) | 16 | 15 | **18** | **4** | 3 | 4 | 14 | 11 | 12 |
| childsnack (20) | 2 | 2 | **3** | 0 | 0 | 0 | 2 | 2 | **3** |
| depot (22) | 10 | 10 | **11** | 4 | 4 | 4 | 10 | 10 | 10 |
| driverlog (20) | 13 | 13 | 13 | 10 | **11** | **11** | 13 | 13 | 13 |
| elevators (50) | 35 | **39** | 37 | 10 | 12 | **14** | **41** | 34 | 33 |
| floortile (40) | 17 | 16 | **19** | 26 | 25 | **27** | 26 | 22 | **27** |
| freecell (80) | 68 | 68 | 68 | 27 | **28** | 27 | **67** | **67** | 66 |
| ged (20) | 15 | 15 | 15 | 10 | 10 | **13** | **19** | **19** | 15 |
| logistics (63) | 28 | 28 | 28 | 19 | **24** | 23 | 28 | 28 | 28 |
| nomystery (20) | 18 | 18 | 18 | **14** | 13 | 14 | 19 | 19 | 19 |
| openstacks (100) | **90** | 89 | 86 | 73 | **75** | 73 | 89 | 88 | 86 |
| parking (40) | 13 | 13 | 13 | 6 | 6 | 6 | 12 | 12 | **13** |
| pegsol (50) | 48 | 48 | 48 | 30 | 31 | **34** | 48 | 48 | 46 |
| petri-net-align (20) | **11** | **11** | 10 | 1 | 1 | 1 | **8** | 6 | 5 |
| pipesw-notank (50) | 24 | 24 | 24 | 9 | 9 | **10** | 24 | 24 | 24 |
| pipesw-tank (50) | **21** | 20 | 18 | 7 | 7 | 7 | **20** | **20** | 19 |
| quantum-layout (20) | 14 | 14 | **15** | 13 | 13 | 13 | 14 | 14 | **15** |
| ricochet-robots (20) | **2** | 1 | 1 | 0 | 0 | 0 | **2** | 1 | 2 |
| rovers (40) | **14** | 13 | 14 | **10** | 9 | **10** | 14 | 14 | 14 |
| slitherlink (20) | 5 | **6** | 5 | 0 | 0 | 0 | 5 | **6** | 5 |
| snake (20) | 8 | **11** | **11** | 0 | 0 | 0 | **8** | 7 | 7 |
| sokoban (50) | 50 | 50 | 50 | 38 | 37 | **38** | 50 | 50 | 50 |
| termes (20) | **12** | 10 | 11 | 7 | **13** | **13** | 13 | 13 | 13 |
| tidybot (40) | **34** | **34** | 32 | 8 | 8 | **10** | 30 | 30 | **32** |
| tpp (30) | 12 | 12 | 12 | **8** | 7 | **8** | 12 | 12 | 12 |
| visitall (40) | 22 | 22 | 22 | 18 | 18 | **19** | 22 | 22 | **23** |
| woodworking (50) | 46 | **48** | 44 | 34 | **46** | 40 | 46 | **49** | 48 |
| others (757) | 469 | 469 | 469 | 342 | 342 | 342 | 449 | 449 | 449 |
| Σ (1806) | **1136** | 1135 | 1133 | 732 | 752 | **762** | **1123** | 1102 | 1105 |

Table 2: Coverage for symbolic search.

get is plenty for A⋆ with potential heuristics to conduct additional analysis to find improving conjunctions. Considering we are typically able to test thousands of different conjunctions within 5 minutes, these results suggest we could obtain better conjunctions by just slightly modifying Algorithm 1 so that it prioritizes more promising conjunctions instead of blindly trying all of them one by one.

The comparison of coverage of symbolic search configurations in Table 2 paints quite different picture from the explicit-state search. Planning directly in Π^$\mathcal{C}$ or Π^$\mathcal{C}_{exact}$ does not seem to have significant effect on $\overrightarrow{A}$ overall although there are few domains where it is beneficial, and it seems to be mostly detrimental for $\overrightarrow{A}$ - $\overleftarrow{I}$. The reason is that planning directly in Π^$\mathcal{C}$ and Π^$\mathcal{C}_{exact}$ tends to consume more memory because all binary decision diagrams (BDDs) grow as the task grows, therefore also manipulating BDDs is slower. In some tasks, there is also an issue with the ordering of BDD variables (corresponding to facts in the task) causing significant blow-up of the BDD encodings. This, unfortunately, makes it harder to apply potential heuristics over conjunction in this setting, because we would need to consider conjunctions that not only improve the informativeness of the heuristics, but also induce a good ordering of BDD variables.

Nevertheless, at least the symbolic backward search $\overleftarrow{I}$ seems to benefit from the compilations (more often from Π^$\mathcal{C}_{exact}$ than from Π^$\mathcal{C}$). The main reason is that the backward search usually generates a smaller number of BDDs representing sets of states than the forward direction, but they typically encapsulate larger number of states. So, it seems having more informative heuristics is able to push this method a little bit further, but enough to see a difference in coverage.

## 9 Conclusion and Future Work

In this work, we focus on higher-dimensional potential heuristics computed via a small number of conjunctions explicitly represented as facts in compilations of the input planning tasks. We propose to use two variants of compilations: a well known Π^$\mathcal{C}$ compilation (Haslum 2012), and a newly introduced Π^$\mathcal{C}_{exact}$ compilation. We compute potential heuristics over conjunctions as single-fact potential heuristics in the compilations. Instead of trying to figure out how to find "good" conjunctions guaranteeing increased informativeness of the resulting heuristics, we focus on a basic question whether heuristic estimates can be improved with just few conjunctions and without significantly increasing computational cost in practice. To this end, we use a very simple greedy algorithm that blindly tries small conjunctions one by one, accepting the conjunction if its addition increases h-value for the initial state. We conclude that even in this simple setting, we can, indeed, increase informativeness of potential heuristics with just few conjunctions. Moreover, we provide a machinery for computing potential heuristics over conjunctions requiring only conjunctions to be plugged-in.

We leave many questions unanswered for future work. The first one is how to determine which conjunctions lead to an improvement. Possible directions for answering this question might be previous works on the selection of improving conjunctions for delete-relaxed heuristics (Fickert and Hoffmann 2017) or for learning no-goods using state-equation heuristics (Steinmetz and Hoffmann 2018). Selection of patterns for pattern databases might also be relevant (Edelkamp 2006; Haslum et al. 2007; Franco et al. 2017; Rovner, Sievers, and Helmert 2019). Another question is the exact relationship between Π^$\mathcal{C}$ and Π^$\mathcal{C}_{exact}$ compilations with respect to the possible potential heuristics that they can express via linear programs (LPs). It would also be interesting to know their relationship to the direct encoding in LP described by Pommerening, Helmert, and Bonet (2017). Lastly, it is not entirely clear how to successfully transfer the increased informativeness of potential heuristics to operator-potential heuristics in the context of symbolic search (Fišer, Torralba, and Hoffmann 2022a,b).

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
