# OpenReview forum: "Towards Feasible Higher-Dimensional Potential Heuristics"
_icaps-conference.org/ICAPS/2024/Conference — ICAPS 2024_

### Official Review · Reviewer_ux57 · 2024-01-20

**Significance And Importance:** 2
**Soundness:** 3
**Novelty:** 3
**Clarity:** 3
**Overall Evaluation:** 1
**Confidence:** 4

**Weaknesses:**

1: Minor weaknesses that are easily fixable.

**Contributions Of The Paper:**

The paper contributes with a new compilation over conjunction of facts in Finite Domain Representation for computing more informative potential heuristics. It also proves the new compilation preserves the reachable state-space, paths and costs of the original planning tasks. Then, the new and a previous compilation are evaluated over several planning tasks using A* and symbolic search in many different settings.

**Ethical Considerations:**

(1) Not Applicable: The paper does not have any ethical considerations to address

**Nomination For Best Paper:**

No

**Questions For Authors:**

Q1. Lines 175-177, does this make each operator preconditions to be non-empty? i.e. if they are originally empty, do you need some kind of extra dummy precondition=
Q2. Lines 295, "facts from X" or "conjunctions from X"?
Q3. Is it correct to say the set of mutexes M is approximated? how hard is to compute that set (i.e. is the complexity of Alg. 1)? how hard is to check the membership of a partial state into that set?
Q4. Fig. 1, does this mean that \Pi^C_{exact} computes higher potential heuristic values in the initial states than \Pi^C, and \Pi^C computes higher values than \Pi?

**Reproducibility:**

4: Authors promise to release code and domains (whichever apply).

**Strengths Of The Paper:**

The main strength of the paper is the theoretical contribution for proving the existence of superfluous paths in the \Pi^C compilation, and the proves regarding with the \Pi^C_{exact} compilation and how it guarantees an isomorphic reachable state-space, paths and costs.

**Weaknesses Of The Paper:**

The major weaknesses of the paper are related to the clarity of the notation (there are multiple dependency levels in the notation with almost no examples, which makes it hard to follow and do the proof-check), and also regarding the experiments, where the compilations do not seem to contribute much for coverage (however, they contribute to the informativeness of the computed heuristic).

Parts of the paper that I found hard to follow:
- Definition 4, the point (4)
- Equations V^{f,0} and expre(o,f,p)
- Second half of the proof of proposition 11

Some other minor details:
- After partial state def., when using |...| explain that denotes the number of facts.
- Sometimes, the letter p is used as a proposition, while in this work is a partial state, this could be confusing.
- Line 159, this line is true only if M represents the set of all possible mutexes (a comment could help, otherwise prove it).
- Line 603, "restart" -> "repeat"?
- Figure 1 caption is not very clear.
- Table 2 focuses on symbolic search with operator potentials but they are shallowly explained in the paper.

---

> ### Author Rebuttal · Authors · 2024-01-27
>
> Thank you for your review.
>
> Q1. No, we allow empty preconditions and goals. The disambiguation of any
> variable v for the empty partial state (empty sets of facts) is simply F_v, thus
> not empty, unless we know that some value of v is not set in *any* reachable
> state. In that case, however, we could also just reduce the domain of v and run
> the compilation on the reduced task. We will make this clear.
>
> Q2. It should be "facts from the conjunctions c \in X", i.e., these are facts
> contained in conjunctions X that are not affected by the operator.
>
> Q3. Deciding whether any given set of facts is mutex is PSPACE-complete in
> general. As in prior work, we restrict ourselves to fact-pair mutexes, and infer
> them from h^2 (Haslum and Geffner AIPS2000), computed once in a pre-process.
> Afterwards, checking whether a pair of facts is mutex is constant time (table
> lookup), and checking whether a partial state contains a (pair) mutex (and is
> therefore mutex itself) is quadratic in the number of variables. Note that the
> (huge) mutex-set \cal{M} does not have to be constructed explicitly at any point.
> We only need to keep the table of mutex pairs; membership in \cal{M} is
> tested as described.
>
> Q4. Yes. However, the comparison between \Pi^C_exact and \Pi^C (right-most
> figure) is done on the same set of conjunctions (in each task) so that we
> compare the effect of compilations rather than the size/quality of conjunctions
> obtained via Alg. 1. We will clarify this.

---

### Official Review · Reviewer_kniL · 2024-01-22

**Significance And Importance:** 2
**Soundness:** 3
**Novelty:** 2
**Clarity:** 2
**Overall Evaluation:** 1
**Confidence:** 3

**Weaknesses:**

0: Minor weaknesses requiring some work to be addressed for the paper to be accepted.

**Contributions Of The Paper:**

This paper addresses the problem of computing potential heuristics over sets of conjunctions via compilation. The proposed approach encodes a set of conjunctions C into single facts in the input problem \Pi and computes single-fact potential heuristics on the resulting compiled problem.

The paper considers the existing exponential compilation \Pi^C to encode a set of conjunctions C and proposes an optimization that uses the notion of mutexes to reduce the overhead. Then, the authors introduce a new compilation  \Pi^C_exact that, differently from \Pi^C, preserves the state space of the original problem \Pi. Lastly, the paper proposes a simple algorithm for computing a set C of conjunctions to improve the heuristic estimates.

The experimental analysis shows that the approach leads to more informative heuristics.

**Ethical Considerations:**

(1) Not Applicable: The paper does not have any ethical considerations to address

**Nomination For Best Paper:**

No

**Questions For Authors:**

1) Can you clarify how the two portfolios P^C and P^C_exact work?

2) What are the differences between the mutex pruning you propose and the mutex pruning adopted by Keyder, Hoffmann, and Haslum (2014)?

3) Keyder, Hoffmann, and Haslum (2014) also point out that there exists a linear encoding that uses conditional effects for encoding the set of conjunctions. Would it be possible to adopt this encoding to compute potential heuristics? Maybe by compiling away conditional effects using the polynomial compilation by Nebel (2000) (On the Compilability and Expressive Power of Propositional Planning Formalisms)

**Reproducibility:**

4: Authors promise to release code and domains (whichever apply).

**Strengths Of The Paper:**

I think that using a compiled representation of the original problem to compute a more informative heuristic is an interesting idea.

The paper is rigorous and contains proofs for theoretical results.
Although I was not able to check all the details, I believe the paper is sound.

The experimental analysis is extensive and proves that the approach leads to moderate improvements with explicit-state search algorithms.

**Weaknesses Of The Paper:**

Clarity - The main weakness of this paper is related to the clarity of the two compilations.
The description of the two encodings is hard to follow, and I could only understand the general idea of the compilation. This is partially because I’m not familiar with the FDR representation. However, I believe that the presentation would benefit from a couple of revisions:
    - I would separate the base compilation from the optimization that uses mutex sets.
    - I suggest adding a complete example of the compilation (e.g., a simple example with a couple of conjunctions and one operator)

Regarding the contributions, it is not clear the relation between the mutex pruning used in this paper and the mutex pruning used in previous works (see question 2).

Novelty and significance - the contributions seem a bit incremental, given that \Pi^C already exists, and the practical usefulness of \Pi^C_exact is not outstanding in terms of coverage improvement.

---

> ### Author Rebuttal · Authors · 2024-01-27
>
> Thank you for your review.
>
> To clarify how the compilations work, we will provide full examples in
> the final version (we either buy an extra page or extend technical report).
>
> 1) Portfolios start by running search with the potential heuristic computed
> in the original task, i.e., the baseline method. If the search runs out of
> memory before finding a plan, then we use the remainder of the time budget to
> run search with potential heuristic computed in \Pi^C (or \Pi^C_exact): first we
> run inference of C for 5 minutes and then run search with heuristic from \Pi^C
> (or \Pi^C_exact). We will make this clear in the final version of the paper.
>
> The point here is that the search with potential heuristics typically terminates
> quickly because it either finds a plan or runs out of memory (that's because
> the computation of h-value for a state is extremely fast so the speed of state
> expansion is similar to blind search). Therefore, with 30 minutes time limit we
> can afford to try running the baseline first before we proceed with more
> elaborate variants.
>
>
> 2) The work by Keyder et al lacks a clear description of how mutex information
> is used exactly, but, based on a code inspection, they follow the same idea
> as we do: using mutexes to discard operators with mutex (context-augmented)
> preconditions. We formalize this idea and analyze the properties of the
> resulting mutex-reduced compilation (which is necessary since, as pointed
> out, desired properties like admissibility are preserved only for the
> reachable states).
>
> Note that the definition does not get much less complicated without mutexes,
> because FDR still requires that we don't allow invalid (partial) states. For
> example, Steinmetz and Hoffmann (2018) needed to use notions of compatibility of
> partial states and regression to achieve that.
>
> 3) This is a very interesting idea worth exploring in future work. Note,
> however, that while Nebel shows that compiling away conditional effects can
> be done without exponential blow-up in task size, this comes at the price of
> a polynomial plan-length increase (due to the introduction of auxiliary
> operators). It is thus not clear immediately how the properties of
> admissibility and consistency within that compilation relate back to the
> original task, i.e., whether an admissible and consistent potential heuristic
> constructed in the compilation can be translated back to an admissible and
> consistent potential heuristic for the original task.

---

### Official Review · Reviewer_rTtn · 2024-01-22

**Significance And Importance:** 2
**Soundness:** 4
**Novelty:** 3
**Clarity:** 3
**Overall Evaluation:** 2
**Confidence:** 5

**Weaknesses:**

1: Minor weaknesses that are easily fixable.

**Contributions Of The Paper:**

The paper introduces a variant of the Pi^C compilation that can track the truth value of conjunctions of facts (C) and in contrast to the original Pi^C compilation has a reachable state space isomorphic to the original task. The authors argue that this is useful for finding higher-dimensional potential heuristics by encoding the higher-dimensional features into the task as individual variables. A simple greedy strategy for finding features proves useful in practice and shows that atomic potential heuristics can be improved by adding a small amount of higher-dimensional (but still small) features.

**Ethical Considerations:**

(1) Not Applicable: The paper does not have any ethical considerations to address

**Nomination For Best Paper:**

No

**Questions For Authors:**

1. Should the proofs for Theorems 7 and 12 also consider reachable but dead states? If so, what is the argument that consistency/admissibility is maintained for them? If not, why is this covered by the existing proofs?
2. It would be interesting to compare the size of generated conjunctions on tasks where coverage improved to the average on all tasks. If those tasks are at the high end of the global distribution, this would suggest that generally, we would like larger/more conjunctions but can only afford to compute enough of them on those tasks. If they are distributed in roughly the same way as the global distribution, it would confirm that small conjunctions are sufficient.

Post rebuttal:
Thank your for your answers. I agree with the updated proof and raised my score for soundness.

**Reproducibility:**

4: Authors promise to release code and domains (whichever apply).

**Strengths Of The Paper:**

The paper is very well written and the proofs are rigorous while still having the right amount of explanations that they remain easy to follow. The main idea is an extension of an existing idea by Steinmetz and Hoffmann who already show the potential heuristics can be defined on the Pi^C compilation. While not groundbreaking, the paper does a good job of showing the problems with this approach and fixing them with the Pi^C_exact compilation.

The empirical evaluation is well-designed, answers interesting questions, and produces state-of-the-art results.

**Weaknesses Of The Paper:**

I have a some minor points to mention here but all are easily fixable. The main one is about a possible gap in two proofs; the remaining ones are about improving clarity, and some related work. If the proofs are correct or can be fixed, the other points are not that critical. I reduced my score for soundness until this point can be cleared up, even if the description of the second-highest score doesn't really reflect my opinion.

Proofs for Theorems 7 and 12, rely on the fact that there is a bijection between plans and intermediate steps. (Side note: in the proof of Theorem 12, the function f_s is never used.) I don't think this is sufficient (at least for consistency) as consistency also has to hold between reachable dead ends which cannot lie on plans. Pi^C can also introduce detours, and I'm not quite convinced that properties hold along those detours. (They probably are as the detour is part of some other plan for which we again can make the original argument but it would be good to spell this out in the paper.)

In the section on the exact Pi^C compilation, it is hard to follow the case distinction of whether a conjunction is added, removed, or stays true/false. In all 4 cases, some of the conditions mentioned are necessary, some sufficient, and some both necessary and sufficient. Making this more explicit would improve readability.

The terms "admissible", "goal-aware" and "consistent" are re-defined to require the property only for reachable states. I don't think this is good practice, as in some contexts, the distinction between being admissible and admissible on the reachable states is important (e.g., orbit search). I would prefer defining a different term over redefining the existing one. Sievers and Helmert (JAIR 2021) use "forward-consistent" and "forward-admissible" for these concepts. Likewise, the term "dead-end" is used in different ways in the literature. Another common interpretation is that dead-ends are states without direct successors.

In Definition 4 the condition v_c \notin V is unclear. Variables v_c should never be in V, i.e., the names "v_c" should be new. Otherwise, it is not clear if the existing variable has the correct semantics.

The observation that a small number of small conjunctions can improve a potential heuristic by a lot has been made before for perfect heuristics (Corrêa and Pommerening ICAPS 2019). I don't think this limits the novelty of the paper as their approach was different and they did not use the resulting heuristics in practice, but it should be discussed as related work.


Minor comments
- line 77 and 78: dom(V) -> dom(v)
- line 80: sets of variables V
- line 164: a fact from X (saying "one" is also true but emphasizes the wrong aspect)
- line 184: missing math mode
- line 212: singular-plural mismatch
- line 243-250: The sentence is too long. Also, since the sentence start with saying you follow up on Steinmetz and Hoffmann's work, it would be more interesting what they used, rather than saying what Haslum used.
- line 279: the opposite implies that o is inapplicable, not that c is false.
- line 303: where only c is true, only c' is true, and where both are true.
- line 335: same length and cost
- line 408: If eff(o) changes a variable from c away from c, then v_c is certainly 0 afterwards but that doesn't necessarily mean it had to be 1 before.
- line 497-478: doesn't this depend on p?
- line 510: let us assume
- line 538: an admissible
- line 555: the Pi^C or the Pi^C_exact compilation
- line 559: running the search
- line 560: a heuristic
- line 567: the reachable state space
- line 590: The next question
- line 642: tends to

---

> ### Author Rebuttal · Authors · 2024-01-27
>
> Thank you for your review.
>
> 1. Indeed, reachable dead ends must also be considered. Thm 12 considers them
> implicitly as its proof is based on isomorphism of state spaces, but the
> proof of Thm 7 should address them explicitly. Nevertheless, both theorems
> hold also for dead ends. We'll correct this in the final version accordingly.
>
> Thm 7:
> Intuitively, if we understand reachable state space S(\Pi) of \Pi as a digraph
> where nodes are reachable states and edges are transitions, then S(\Pi) is a
> subgraph of S(\Pi^C), which implies that admissibilty and consisency are
> preserved. Prop. 5 then shows that S(\Pi^C) does not introduce any shortcuts to
> its subgraph S(\Pi).
>
> Formally, we can prove consistency by contradiction.
> Let h^C, h be as stated in Thm 7, and suppose we have reachable states s,s' and
> an operator o in \Pi s.t. o[s] = s' and h(s) > h(s') + c(o).
> From Prop. 6 we have that both C[s] and C[s'] are reachable in \Pi^C and there
> is an operator o^X s.t. o^X[C[s]] = C[s']. Therefore we have
> h^C(C[s]) > h^C(C[s']) + c(o), and from consistency of h^C in \Pi^C, we have
> h^C(C[s']) + c(o^X) >= h^C(C[s]) > h^C(C[s']) + c(o), which is contradiction
> because c(o^X) = c(o) by definition.
>
> For admisibility, assume a reachable state s s.t. h(s) > h^\*(s).
> From Prop. 6: h^C(C[s]) > h^\*(s).
> From admissibility of h^C: h^\*C(C[s]) >= h^C(C[s]) > h^\*(s) where h^\*C is an
> optimal heuristic for \Pi^C. Therefore there is a cheaper path to a goal from
> s in \Pi then from C[s] in \Pi^C (or a path from s exists but not a path from
> C[s]), which is a contradiction, because Prop. 6 says that if there was such
> path in \Pi it would also be in \Pi^C. (By Prop. 5 we also have h^*C(C[s])=h^\*(s).)
>
> Note that neither argument above require a plan but only reachability via some
> path so it generalizes over dead ends.
>
> Thm 12:
>
> (A) same argument as for Thm 7 except we use Prop. 11 instead of 6.
>
> (B) is similar to (A) except we use Prop. 9 and Prop. 10 to show that for every
> path in \Pi there is exactly one path in \Pi^C_exact.
>
>
> 2. It sounds interesting and we'll try to come up with some way to include this
> analysis.
>
> We'll use forward admissible (consistent) instead of admissible (consistent).
> We agree it'll make it more clear.
>
> Thank you for pointing out the work of Correa and Pommerening (2019). We're
> actually aware of their work and had it in our draft, but it somehow
> slipped away when shortening the paper for submission. We'll definitely include
> a discussion of their work.

---

### Meta-Review · Area_Chair_hA41 · 2024-02-06

**Recommendation:** Accept (Oral)
**Confidence:** 5

**Metareview:**

The paper computes potential heuristics over conjunctive features via compilation. For this purpose, the authors consider a variation of the $\Pi^C$ compilation ($\Pi^C_\textup{exact}$) that guarantees an isomorphic reachable state-space. The reviewers are happy with the quality, originality and significance of the work.

One reviewer had a question regarding the soundness of one proof that was satisfactory resolved in the rebuttal.

Pros:
- interesting contribution
- paper is rigorous and mostly well-written
- experiments are well-designed

Cons:
- some parts of the paper hard to follow
- $\Pi^C_\textup{exact}$ compilation is somewhat incremental (based on ideas behind $\Pi^C$) and does not have an outstanding experimental impact

**Ethical Considerations:**

(1) Not Applicable: The paper does not have any ethical considerations to address